# Analyzing the Confidentiality of Undistillable Teachers in Knowledge Distillation

**Souvik Kundu, Qirui Sun, Yao Fu, Massoud Pedram, Peter A. Beerel**
Electrical and Computer Engineering
University of Southern California
Los Angeles, CA 90089
`{souvikku, qiruisun, yaof, pedram, pabeerel}@usc.edu`

## Abstract

Knowledge distillation (KD) has recently been identified as a method that can unintentionally leak private information regarding the details of a teacher model to an unauthorized student. Recent research in developing undistillable nasty teachers that can protect model confidentiality has gained significant attention. However, the level of protection these nasty models offer has been largely untested. In this paper, we show that transferring knowledge to a shallow sub-section of a student can largely reduce a teacher's influence. By exploring the depth of the shallow subsection, we then present a distillation technique that enables a skeptical student model to learn even from a nasty teacher. To evaluate the efficacy of our skeptical students, we conducted experiments with several models with KD under both training data-available and data-free scenarios for various datasets. While distilling from nasty teachers, compared to the normal student models, skeptical students consistently provide superior classification performance of up to $\sim 59.5\%$. Moreover, similar to normal students, skeptical students maintain high classification accuracy when distilled from a normal teacher, showing their efficacy irrespective of the teacher being nasty or not. We believe the ability of skeptical students to largely diminish the KD-immunity of a potentially nasty teacher will motivate the research community to create more robust mechanisms for model confidentiality. We have open-sourced the code at `github.com/ksouvik52/Skeptical2021`.

## 1 Introduction

Knowledge distillation (KD) [10] aims to transfer the useful knowledge of a trained model (the teacher) to another model (the student). KD has found success in various applications [1, 29, 3, 17] and is particularly useful for resource-constrained IoT applications where the compute budget is limited and compute-efficient models are required. Generally, KD requires the student model to be trained over the same data-set that is used to train the teacher. However, recent research [31, 20, 2] has shown the efficacy of KD even under the "data-free" scenario where the training data may not be available for the student to get trained.

Over the past few years various forms of distillation have been proposed, including distillation from the student itself [36] and via an ensemble of students [37]. However, recently, reference [18] has highlighted the fact that many forms of distillation may unintentionally leak the IP of a teacher model. In particular, some teacher models contain significant IP associated with the arduous effort of both data collection and model training, which motivates their release as "black-box" executable pieces of software. Moreover, these trained models may even enable safeguarding the training data (for example, sensitive medical images [28] and company proprietary information [27, 30]) as well as their performance on the secure data. Use of KD under these circumstances, sometimes

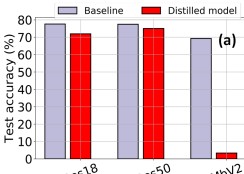 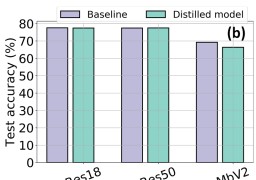 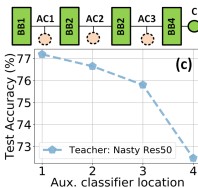

Figure 1: Distillation from a nasty ResNet50 to (a) normal students, (b) proposed skeptical students, on CIFAR-100. In particular, for MobileNetV2 (MbV2) which is a reduced parameter model, the proposed distillation method can improve the accuracy by $59.49\%$. (c) Impact of transferring knowledge at various depth of a ResNet18 from a nasty teacher. BB represents a basic-block layer.

under the "data-free" condition, may enable training of an unauthorized student to yield comparable performance as the teacher. To mitigate this issue, reference [18] has proposed the idea of a *nasty teacher* that prevents knowledge leaking to a student and thereby reduces the student's classification performance. Fig. 1(a) depicts the the success of a nasty ResNet50 in degrading various student models' classification performances compared to their respective baseline performances[1].

**Our contributions**: Earlier evaluations [18] have shown that nasty teachers can retain their efficacy under various settings of two key hyperparameters, namely the weight of the distillation loss ($\alpha$) and the softmax temperature ($\tau$) of the distillation loss. However, a comprehensive evaluation of the efficacy of *undistillable* nasty teachers has yet to be completed. Towards this goal, we investigate the performance of KD on distillation at different depths of the student model, and in particular, how the teacher's influence changes when transferring knowledge to an intermediate shallow section of the student through an auxiliary classifier (AC). We find that the impact of the nasty teacher drastically reduces as we transfer knowledge to a shallow subsection of the student (Fig. 1(c)).

Based on these findings, we present a *skeptical student* that uses an intermediate shallow auxiliary classifier to transfer the information derived through the soft probabilities of the teacher's output classes. We further propose a novel *hybrid distillation* scheme to improve learnability of the student by distilling both from a teacher and the student itself. Our approach has some similarity with self-distillation (SD) [36] because both approaches use an auxiliary classifier for the knowledge transfer. However, the goal of SD is to show the efficacy of a model distilling from itself, contrasting our goal of analyzing the possible presence of a potential model stealer who can extract knowledge even from an undistillable nasty teacher. More importantly, the proposed hybrid distillation is effective in stealing a model's IP even under a "data-free" scenario, contrasting SD which is only applicable for students who have access to training data.

We conduct extensive experiments using both standard KD with available training data on CIFAR-10, CIFAR-100, and Tiny-ImageNet, and data-free KD on CIFAR-10 testset. Experimental results show that compared to normal ones, skeptical students exhibit improved performance of up to $\sim 59.5\%$ and $\sim 5.8\%$ for data-available and data-free KD, respectively, when distilled from nasty teachers. This exposes a significant limitation of nasty teachers attempting to protect model IP. Moreover, our proposed students perform similar to normal student models while distilled from normal teachers, demonstrating their efficacy irrespective of the teacher being nasty or not.

## 2 Preliminaries

### 2.1 Knowledge Distillation

Knowledge distillation, similar to the goal of various model compression techniques [13, 7, 5, 14, 16], embeds the rich information of a compute-heavy model into a model that generally requires fewer computations. The traditional KD [10] relies on information transfer through a Kullback–Leibler (KL) divergence measure between the soft logits at the output classifier layers of the teacher and the student. Apart from this, over the past few years, various other efficient distillation methods have been proposed, including distilling from hints provided by the teacher [25], distilling via attention transfer [34], and other approaches [23, 21]. To reduce the distillation-based training time and avoid the requirement of a separate teacher, reference [36] has proposed self-distillation. The authors partition

---

[1]The baseline models are trained with only cross-entropy (CE) loss.

Table 1: Performance of student (ResNet18) under transferability test on CIFAR-100.

| Teacher | Teacher type | Teacher Acc % | Student Acc % | $\Delta_{base}$ |
|---------|--------------|---------------|---------------|-----------------|
| ResNet50 | Nasty | 76.57 | 72.47 | -5.08 |
| ResNet18 | Distilled | 72.47 | 70.99 | -6.56 |
| ResNet50 | Normal | 78.04 | 79.39 | +1.84 |
| ResNet18 | Distilled | 79.39 | 79.47 | +1.92 |

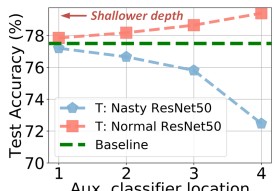

Figure 2: A ResNet18 student's performance on CIFAR-100 dataset.

the student model into several shallow sections, each having its own auxiliary classifier, to which the final classifier transfers the soft-logits to enhance the model's classification performance. Several recent studies [32, 33] have also shown the efficacy of KD to a student from its own pre-trained variant as its teacher. Similar to [18], in this work, we assume to have no access to the teacher model's intermediate features. Rather, we focus on the information leaking to an unauthorized outsider and thus rely on standard KD.

## 2.2 Model IP protection

Protection of model IP has drawn significant interest primarily due to the massive human resource and financial costs required for large model training. Earlier works explored various defense strategies, including adaptive misinformation against model performance stealing [11] and passport-based defense [35]. In most of these cases, the stealer is assumed to have access to only synthetically generated data. However, also of interest is when a portion of training data is unintentionally leaked.

## 2.3 Poisoning of Neural Network Models

An attacker can degrade the performance of a neural network by simply injecting poisoned data [22] into the training set. Adversarial-attack generated images [6, 19, 15] have proven to be effective in degrading a model's performance. Backdoor attacks [4] insert crafted malicious data into the training set that apparently trains the model to perform well until such time that the attacker sets up a signal that degrades the model performance drastically. The Bit flip attack [24] corrupts selective bits of the trained DNN weights to lower its performance. As mentioned earlier, reference [18] has recently proposed to poison a neural network model through training, such that the model retains its classification performance, but loses its ability to be used as an efficient teacher, thus referred to as a nasty teacher. These nasty teachers are believed to protect the model IP.[2] This work analyzes the degree of IP protection that such teachers provide and, in particular, presents a skeptical student-based distillation technique that diminishes the effect of their nastiness, as detailed in the next Section.

## 3 Motivational Case Study

To motivate our skeptical students, this section presents an empirical analysis that explore the efficacy of nasty teacher models under two distinct KD scenarios[3].

### 3.1 Transferability of the Impact of Nasty Teachers

**Definition 1.** *Secondary student:* We define a secondary student as a model that is trained via KD from a trained model which was earlier trained via distillation from a teacher model. In this context, we refer to the student that is distilled from the original teacher model, as the *primary student*. We measure the *transferability impact (TI) of a teacher* as the performance improvement (or degradation) of a secondary student with respect to its baseline ($\Delta_{base}$). A negative $\Delta_{base}$ signifies the success of a teacher's privacy or confidentiality preserving effort [18].

---

[2]In so far as a teacher is trying to protect its private IP from inquiring or even intrusive students, we believe a better phrase to characterize him/her is as a "defensive teacher" or in the worst case as a "secretive teacher". In our view, there is nothing "nasty" about what the teacher is attempting to do. However, because the prior art has consistently used the term "nasty teacher," we will also use that phrase in this paper.

[3]The description of training hyperparameters is given in Section 5.1 for all the experiments in this section.

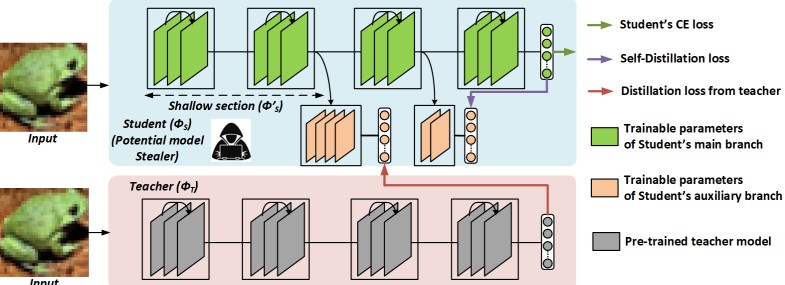

Figure 3: Skeptical student distillation framework. Note the arrow of the distillation loss components are directed from teacher to student for the corresponding KL-divergence computation.

We first trained both normal as well as nasty variants of a ResNet50 on CIFAR-100. We then used two ResNet18 students to distill knowledge from these two teachers. Finally, to test TI we used the distilled ResNet18 models as teachers to secondary students with identical model architectures.

**Observation 1.** *The TI of a nasty teacher is negative, meaning the nastiness of a teacher transfers to its student.*

Table 1 shows the classification performance of a secondary student is $6.56\%$ lower than the baseline while using a distilled ResNet18 as teacher. Interestingly, this implies that the false sense of generalization that the primary student inherits makes it become a nasty teacher. Thus simple redistillation from a primary to a secondary student will not evade the nastiness of a teacher. [4]

### 3.2  Transferring Knowledge to a Shallow Subsection of the Student

We computed the weighted KL-divergence based distillation loss at an auxiliary classifier (AC) of the student, and used its final classifier (C) to compute the weighted CE-loss only. We evaluated the impact of the auxiliary classifier placed at different depths of the student model. In particular, for the ResNet18 student we performed four experiments by placing the AC after the $n^{th}$ BB, $n$ ranging from 1 to 4, where the model's final classifier is located at the end of $4^{th}$ block. We follow similar procedure as [36] to design the AC branches.

**Observation 2.** *The influence of a teacher (both nasty and normal) on a student's performance reduces as we distill knowledge to a shallow subsection of the student model.*

Fig. 2 shows the accuracy of the student model approaches the baseline accuracy as we distill from the teacher at shallower depths ($n = 1$). Interestingly, this trend can be observed for both nasty and normal teachers. In particular, from a nasty ResNet50, the student ResNet18 has a test accuracy of up to $77.19\%$ while distilled at AC 1, in contrast the accuracy is $72.47\%$ when distilled at the final classifier. On the other hand, ResNet18 can have a classification accuracy reduction of as much as $1.56\%$ when distilled at shallow depths.

We leverage these two observations in our hybrid distillation on the skeptical students and diminish the transferring of teachers' nastiness, presented in the next section.

## 4  Skeptical Students

Let us consider a student model $\Phi_S$, which distills knowledge from a pre-trained teacher $\Phi_T$ where $g_{\Phi_S}(.)$ and $g_{\Phi_T}(.)$ are the functions describing the student and teacher models, respectively. Let ($\boldsymbol{x}$, $\boldsymbol{y}$) be the vectorized pairs of inputs and corresponding output labels used to train these models. For teacher-based traditional KD [10], the training loss for the student may be written as

$$\mathcal{L}_{KD} = (1-\alpha) * \mathcal{L}_{\mathcal{CE}}\big(\sigma(g_{\Phi_S}(\boldsymbol{x},\boldsymbol{y}))\big) + \alpha * \tau^2 * \mathcal{L}_{\mathcal{KL}}\big(\sigma(g_{\Phi_S}(\boldsymbol{x},\boldsymbol{y}),\tau), \sigma(g_{\Phi_T}(\boldsymbol{x},\boldsymbol{y}),\tau)\big) \quad (1)$$

where $\mathcal{L}_{\mathcal{CE}}$ represents the student's cross-entropy (CE) loss and $\mathcal{L}_{\mathcal{KL}}$ represents the KL divergence loss aimed at transferring knowledge. Here, $\sigma(.)$ is the softmax function and $\tau$ is the softmax

---

[4]It may also be interesting to note that, as shown in Table 1, the TI of a normal teacher follows a similar trend and remains positive.

temperature, both of which are used to compute soft probabilities. $\tau$ is set to 1 for the CE loss. The hyperparameter $\alpha$ acts as a balancing factor between the two loss terms. Based on the observations in Section 3, we propose a skeptical student that can largely diminish its nastiness at the very first round of distillation. In particular, skeptical students are those models that always transfer the teacher's KL-div driven knowledge at a shallow section ($\Phi'_S$), as depicted in Fig. 3. Thus the teacher driven loss, $\mathcal{L}_T$ can be formulated same as the $\mathcal{L}_{KD}$ with $\Phi'_S$ replacing $\Phi_S$ in both the CE and KD loss components. To train the complete student model $\Phi_S$ we rely on the CE-loss applied at the final classifier. However, due to reduced influence of the teacher, such students can hardly get any benefit of KD from a normal teacher. Hence, to improve these models' performance, motivated by the idea of self-distillation (SD), we introduce a third loss term that allows distillation to shallow ACs from the student's final classifier,

$$\mathcal{L}_{SD} = \sum_{j \in \mathcal{J}} \left\{ (1 - \beta) * \mathcal{L}_{\mathcal{CE}}\big(\sigma(g_{\Phi_S^j}(\boldsymbol{x}, \boldsymbol{y}))\big) + \beta * \mathcal{L}_{\mathcal{KL}}\big(\sigma(g_{\Phi_S^j}(\boldsymbol{x}, \boldsymbol{y}), \tau), \sigma(g_{\Phi_S}(\boldsymbol{x}, \boldsymbol{y}), \tau)\big) \right\} \quad (2)$$

Note that, $\mathcal{J} \in AC_i$ where, $N > i > i_{\Phi'_S}$[5]. Here $N$ and $i_{\Phi'_S}$ represent the total number of sub blocks in the model and sub-block index at which the AC of $\Phi'_S$ is placed, respectively. Finally, our hybrid distillation loss is given by,

$$\mathcal{L}_S = \gamma_1 \mathcal{L}_T + \gamma_2 \mathcal{L}_{SD} + \gamma_3 \mathcal{L}_{\mathcal{CE}}\big(\sigma(g_{\Phi_S}(\boldsymbol{x}, \boldsymbol{y}))\big) \quad (3)$$

where the last term corresponds to the CE loss of the complete student $\Phi_S$. Here, $\gamma_1, \gamma_2,$ and $\gamma_3$ are hyperparameters to balance the KD loss of the auxiliary classifier, self-distillation loss, and the CE loss of the student. It is noteworthy that we use the auxiliary classifier sections during training and that these auxiliary sections may be removed during inference, nullifying any extra inference parameter cost.

**Skeptical students for data-free KD.** As described earlier, the skeptical students primarily use an intermediate auxiliary classifier to distill knowledge. Therefore, teaching the remaining part of the network ($\Phi_S$ - $\Phi'_S$) is, in particular, difficult for data-free KD because there is no CE-loss to train the whole network $\Phi_S$. To mitigate this issue we propose an *auxiliary self distillation* loss to distill knowledge to the final classifier from the intermediate auxiliary classifier. Note that, as depicted in Fig. 4, the same auxiliary classifier works as a student to learn from a teacher under the data-free scenario. To evaluate this, we use recently proposed zero-shot knowledge transfer [20] with the skeptical students using a loss function enhanced by the auxiliary self KD,

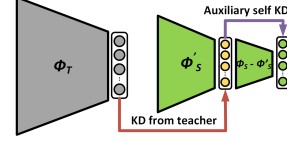

Figure 4: Data-free distillation from a teacher to a skeptical student.

$$\mathcal{L}_{S_{DF}} = \mathcal{L}_{\mathcal{KL}}\big(\sigma(g_{\Phi'_S}(\boldsymbol{x}, \boldsymbol{y}), \tau), \sigma(g_{\Phi_T}(\boldsymbol{x}, \boldsymbol{y}), \tau)\big) + \mathcal{L}_{\mathcal{KL}}\big(\sigma(g_{\Phi_S}(\boldsymbol{x}, \boldsymbol{y}), \tau), \sigma(g_{\Phi'_S}(\boldsymbol{x}, \boldsymbol{y}), \tau)\big) + \gamma_{at} \mathcal{L}_{\mathcal{AT}}$$
$$(4)$$

The first term takes care of knowledge transfer from the teacher, while the second term helps train the final classifier. Similar to the original paper [20], we also use an attention-transfer loss $\mathcal{L}_{\mathcal{AT}}$.

**Nasty teacher training.** As proposed in [18], we use self-undermining knowledge distillation to design the nasty teacher $\Phi_T$. Specifically, we train $\Phi_T$ via distillation from a pre-trained model $\Phi_A$ with the same network architecture, minimizing the following loss

$$\mathcal{L}_N = \mathcal{L}_{\mathcal{CE}}\big(\sigma(g_{\Phi_T}(\boldsymbol{x}, \boldsymbol{y}))\big) - \alpha_N * \tau_N^2 * \mathcal{L}_{\mathcal{KL}}\big(\sigma(g_{\Phi_T}(\boldsymbol{x}, \boldsymbol{y}), \tau_N), \sigma(g_{\Phi_A}(\boldsymbol{x}, \boldsymbol{y}), \tau_N)\big) \quad (5)$$

where the CE loss terms helps retain $\Phi_T$'s classification performance. The second term maximizes the KL divergence between $\Phi_A$ and $\Phi_T$ allowing $\Phi_T$ to learn a false form of generalization that plays a key role in its undistillability [18]. Here, $\tau_N$ is the softmax temperature, similar to traditional KD, and $\alpha_N$ controls the severity of the self-undermining distillation of the nasty teacher.

## 5 Experimental Results

### 5.1 Experimental Setup

**Models and Datasets.** To evaluate the efficacy of our hybrid distillation approach, we performed detailed experiments on three popular datasets, CIFAR-10, CIFAR-100 [12], and Tiny-ImageNet [8]

---

[5]To minimize the auxiliary layer computation overhead during training, in this paper we use 1 AC to transfer the teacher knowledge and 1 AC for SD.

Table 2: Performance of normal vs. skeptical student when distilled from a nasty teacher.

| Dataset | $\Phi_T$ | $\Phi_T$ Acc. (%) | $\Phi_S$ | $\Phi_S$ Base-line Acc. (%) | Student Acc. (%) | | | $\Delta_{acc}$ |
|---|---|---|---|---|---|---|---|---|
| | | | | | Normal ($acc_n$) | Skeptical ($acc_s$) | Skeptical-E ($acc_{s_e}$) | |
| CIFAR-10 | ResNet18 | 94.67 | ResNet18 | 95.15 | 94.13(±0.18) | **95.09**(± 0.15) | 94.77(± 0.05) | +0.96 |
| | | | MobileNetV2 | 90.12 | 88.13(±0.13) | **90.37**(± 0.25) | 90.21(± 0.18) | +2.24 |
| | ResNet50 | 94.28 | ResNet18 | 95.15 | 94.38(±0.18) | **95.16**(± 0.01) | 95.02(± 0.01) | +0.78 |
| | | | ResNet50 | 94.9 | 94.21(±0.04) | **95.48**(± 0.14) | 95.48(± 0.14) | +1.27 |
| | | | MobileNetV2 | 90.12 | 88.76(±0.14) | **91.02**(± 0.09) | 90.88(± 0.23) | +2.26 |
| CIFAR-100 | ResNet18 | 77.55 | ResNet18 | 77.55 | 75.00(±0.14) | **77.33**(± 0.21) | 76.38(± 0.1) | +2.33 |
| | | | MobileNetV2 | 69.24 | 7.13(±0.71) | **66.62**(± 0.30) | 64.26(± 0.64) | +59.49 |
| | ResNet50 | 76.57 | ResNet18 | 77.55 | 72.28(±0.27) | **77.25**(± 0.25) | 75.48(±0.54) | +4.97 |
| | | | ResNet50 | 78.04 | 74.14(±0.85) | **78.65**(± 0.29) | 77.61(±0.1) | +4.52 |
| | | | MobileNetV2 | 69.24 | 7.72(±1.57) | **66.38**(± 0.50) | 62.93(±0.75) | +58.66 |
| Tiny-ImageNet | ResNet18 | 62.08 | ResNet18 | 63.07 | 53.60(±0.04) | **65.76**(±0.83) | 60.63(±0.07) | +12.16 |
| | | | MobileNetV2 | 57.01 | 4.81(± 0.19) | **54.74**(±0.84) | 54.27(±2.94) | +49.93 |

with ResNet18, ResNet50 [9] and MobileNetV2 [26] models. We used PyTorch API to define and train our models on an Nvidia RTX 2080 Ti GPU.

**Training hyperparameters.** We used standard data augmentation techniques (horizontal flip and random crop with reflective padding) and the SGD optimizer for all training. To create a nasty teacher, we first trained a network $\Phi_A$ for 160 epochs on CIFAR-10 and 200 epochs for CIFAR-100 and Tiny-ImageNet with an initial learning rate (LR) of 0.1 for all. For CIFAR-10, we reduced the LR by a factor of 0.1 after 80 and 120 epochs. For CIFAR-100 and Tiny-ImageNet the LR decayed at 60, 120, and 160 epochs by a factor of 0.2. We then trained the nasty $\Phi_T$ of same architecture with the same epochs and LR hyperparameters. We chose $\alpha_N$ as 0.04, 0.005 and 0.005, for CIFAR-10, CIFAR-100, and Tiny-ImageNet, respectively [18]. Similar to [18], we chose $\tau_N$ to be 4, 20, and 20 for the three datasets. For the distillation training to $\Phi_S$ (both normal and skeptical), we trained for 180 epochs with a starting LR of 0.05 that decays by a factor of 0.1 after 120, 150, and 170 epochs. Unless stated otherwise, we kept $\tau$ the same as $\tau_N$ and chose $\alpha$ and $\beta$ to be 0.9 and 0.7, respectively. We placed the skeptical students' auxiliary classifiers after the $2^{nd}$ ($\Phi'_S$ for KD from the teacher) and $3^{rd}$ (for SD) BB of a total of 4 ResNet blocks. To give equal weight to the loss components of Eq. 3, we chose $\gamma_1 = \gamma_2 = \gamma_3 = 1.0$, for all the experiments. We performed all the experiments with two different seeds and report the average accuracy with std deviation (in bracket) in the tables.

## 5.2 Data-available Distillation

To evaluate model performance, we conducted two types of distillation: distill to self (DtoS) [32], where both teacher and student architectures are the same, and KD from a compute heavy teacher to a reduced-parameter student (for example,$\Phi_T$: ResNet50, $\Phi_S$: ResNet18, MobileNetV2). For DtoS, we also performed distillation with both the assumption of the model being heavy ($\Phi_{T/S}$: ResNet50) and lite ($\Phi_{T/S}$: ResNet18). Table 2 shows the corresponding performance when distilled from a nasty teacher. Skeptical students **always** outperform their normal counterparts providing better accuracy with improvements of up to ∼59.49%. *These results clearly show the efficacy of skeptical students in mitigating the undistillability of a nasty teacher*. We also measure the classification performance by ensembling the ACs and final classifier outputs and denote that as 'Skeptical-E'. However, the ensemble performance is always inferior to the final classifier, which is primarily due to inferior performance of the AC that distills knowledge from the nasty teacher.

Table 3 shows the performance of both skeptical and normal students when distilled from a normal teacher. As we can see, the ensemble output of skeptical students perform better than their normal counterparts. *These results motivate the use of of skeptical students for distillation irrespective of whether the teacher is nasty or not.* In both the tables $\Delta acc$ is the accuracy difference between a skeptical and corresponding normal student when both are trained via distillation from a teacher, i.e. $\Delta_{acc} = \{max(acc_s, acc_{s_e}) - acc_n\}$.

Table 3: Performance of normal vs. skeptical student when distilled from a normal teacher.

| Dataset | $\Phi_T$ | $\Phi_T$ Acc. (%) | $\Phi_S$ | $\Phi_S$ Base-line Acc. (%) | Student Acc. (%) | | | $\Delta_{acc}$ |
| | | | | | Normal ($acc_n$) | Skeptical ($acc_s$) | Skeptical-E ($acc_{s_e}$) | |
|---|---|---|---|---|---|---|---|---|
| CIFAR -10 | ResNet18 | 95.15 | ResNet18 | 95.15 | 95.38 ($\pm$0.10) | **95.45**($\pm$0.10) | 95.42($\pm$0.09) | +0.07 |
| | | | MobileNetV2 | 90.12 | 91.36($\pm$0.17) | 91.81($\pm$0.15) | **92.00**($\pm$0.28) | +0.64 |
| | ResNet50 | 94.9 | ResNet18 | 95.15 | **95.43**($\pm$0.11) | 95.31($\pm$0.01) | 95.27($\pm$0.04) | -0.12 |
| | | | ResNet50 | 94.9 | 95.15($\pm$0.13) | 95.85($\pm$0.05) | **96.09**($\pm$0.01) | +0.94 |
| | | | MobileNetV2 | 90.12 | 91.71($\pm$0.06) | 91.71($\pm$0.18) | **91.95**($\pm$0.16) | +0.24 |
| CIFAR -100 | ResNet18 | 77.55 | ResNet18 | 77.55 | 78.96($\pm$0.12) | 78.79($\pm$0.42) | **79.68**($\pm$0.52) | +0.72 |
| | | | MobileNetV2 | 69.24 | 75.12($\pm$0.08) | 71.63($\pm$0.19) | **75.45**($\pm$0.06) | +0.33 |
| | ResNet50 | 78.04 | ResNet18 | 77.55 | 79.21($\pm$0.24) | 78.51($\pm$0.44) | **79.86**($\pm$0.01) | +0.65 |
| | | | ResNet50 | 78.04 | 79.56($\pm$0.13) | 80.66($\pm$0.52) | **81.96**($\pm$0.52) | +2.4 |
| | | | MobileNetV2 | 69.24 | 75.28($\pm$0.04) | 71.76($\pm$0.16) | **76.32**($\pm$0.34) | +1.04 |
| Tiny-ImageNet | ResNet18 | 63.07 | ResNet18 | 63.07 | 67.35($\pm$0.18) | 66.49($\pm$0.30) | **67.43**($\pm$0.47) | +0.08 |
| | | | MobileNetV2 | 57.01 | 64.99($\pm$0.51) | 59.37($\pm$0.01) | **65.38**($\pm$0.01) | +0.39 |

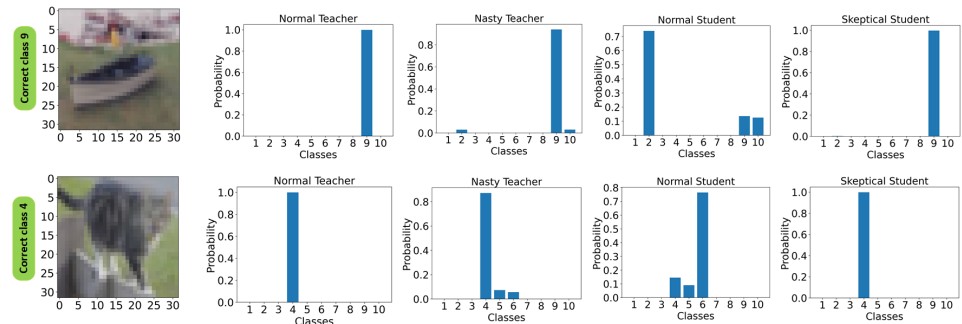

Figure 5: Logit response visualization after the softmax layer. Each row contains an example image from CIFAR-10 dataset and corresponding response for normal teacher, nasty teacher, normal student and skeptical student. We used ResNet50 and ResNet18 as teacher and student model, respectively.

## 5.3 Qualitative Analysis

We now present qualitative behavioral analysis of both normal and skeptical students upon distillation from both normal and nasty teachers. Fig. 5 shows that the nasty teacher has multiple non-negligible peaks at its final softmax logit response, in contrast to the normal teacher having mainly one high valued peak. As mentioned in [18], this can create a false sense of generalization to a normal student causing the student to misclassify, as shown by its logit response. Our skeptical students, on the other hand, not only classify correctly, but also largely mitigate the issue of multi peak logit response of a normal student.

We present visualizations of the t-distributed stochastic neighbor embedding (t-SNE) for output logits in Fig. 6. It shows that the inter class cluster distance is shifted and reduced for certain classes of the nasty teachers. A similar shift of class clusters is also observed for the normal students and even in the AC of the skeptical student where teacher knowledge is transferred. However, the final classifier of the skeptical students has a similar class clustering distribution as a normal teacher. This demonstrates that the remaining sections of the student model ($\Phi_S$ - $\Phi'_S$) indeed remain free from the impact of the nasty teacher.

## 5.4 Ablation Studies

**Ablation study with the Temperature $\tau$.** To further evaluate the influence of the hyperparameter $\tau$ on the student distillation, we performed ablation with $\tau \in [2, 5, 10, 15, 20]$. As depicted in Fig. 7(a), when distilling from a nasty teacher, the skeptical students maintain their superiority compared to their normal counterparts at all different values of $\tau$. While distilling from a normal teacher, even at reduced $\tau$, both the normal and skeptical student variants retain higher classification accuracy compared to their baseline (Fig. 7(c)).

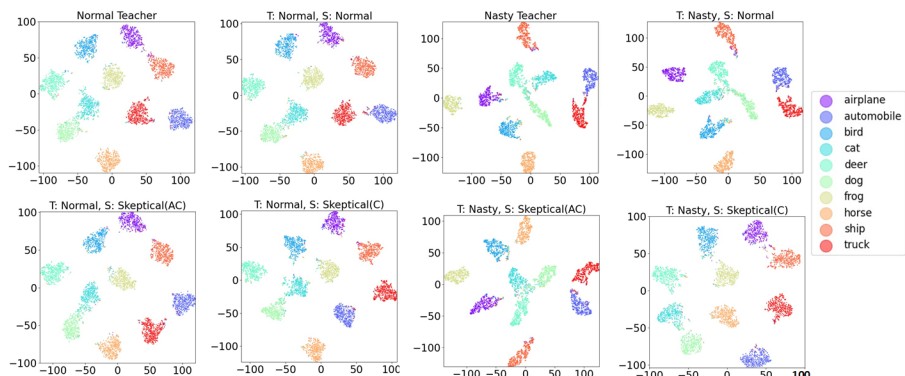

Figure 6: Visualization of tSNE for normal and skeptical students (ResNet18) upon distillation from both normal and evasive teacher (ResNet50) on CIFAR-10. For the skeptical students we plot visualization both at the final classifier (C) and auxiliary classifier (AC).

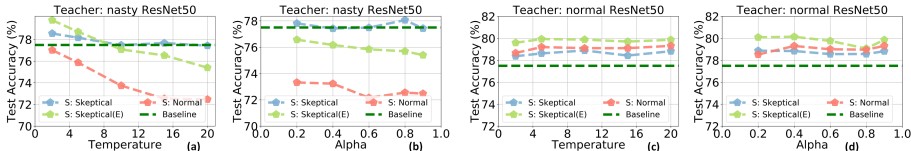

Figure 7: Ablation study with $\alpha$ and $\tau$ for normal and skeptical students (ResNet18) upon distillation from both normal and nasty teacher (ResNet50) on CIFAR-100.

**Ablation study with the balancing term** $\alpha$. To determine the influence of the undistillable teacher on the performance of the presented models, we conducted distillation with $\alpha \in [0.2, 0.4, 0.6, 0.8, 0.9]$. As $\alpha$ reduces, the influence of the teacher is reduced and we see an obvious improvement in student performance. Interestingly, as we can see in Fig. 7(b), *even at reduced $\alpha$ the skeptical students maintain improved performance compared to their normal counterparts.* In distillation from a normal teacher, similar to the previous ablation, the skeptical students do not suffer from any significant performance drop compared to normal students (Fig. 7(d)).

## 5.5 Limited Data and Data-Free Distillation

Instead of having full access to all training samples, KD with limited or no access to training sample is considered a more realistic scenario for model stealing. Fig. 8 shows the students' performance upon distillation from a teacher (both normal and nasty) when only a fraction of the total training data is available. In particular, the figure shows under different % of training data availability the skeptical student performs consistently better than its normal variant upon distillation from a nasty teacher. When distilled from a normal teacher, the skeptical student perform similar to its normal counterpart.

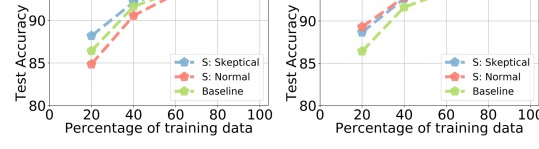

Figure 8: ResNet18 on CIFAR-10 dataset under different percentage of limited training data.

To demonstrate skeptical student's performance under data-free scenario, we leverage the idea of zero shot knowledge transfer [20], a state-of-the-art data-free distillation technique. For this evaluation we used ResNet34 and ResNet50 as teacher models with ResNet18 as the student for both, on CIFAR-10. We used the same training hyperparameters as in [20] with the proposed loss introduced in Eq. 4 and evaluated the performance when the teacher is both grey and white boxed. In particular, for the grey-box and black-box assumptions, we computed the final loss with and without the attention-transfer (AT) loss from the teacher, respectively. Table 4 shows the skeptical students always yield higher classification accuracy compared to their normal counterparts upon distillation from both normal and nasty teachers. In particular, while distilling from a nasty teacher the student's

Table 4: Performance of normal vs. skeptical student on data-free distillation [20] from a teacher.

| Dataset | $\Phi_T$ | $\Phi_T$ type | $\Phi_T$ Acc. (%) | $\Phi_S$ | Student Acc. (%) | | $\Delta_{acc}$ |
|---|---|---|---|---|---|---|---|
| | | | | | Normal | Skeptical | |
| With AT loss (grey-box) | | | | | | | |
| CIFAR -10 | ResNet34 | Nasty | 94.81 | ResNet18 | 87.7($\pm$1.20) | **91.76**($\pm$0.30) | +4.06 |
| | | Normal | 95.3 | | 93.41($\pm$0.21) | **93.52**($\pm$0.06) | +0.11 |
| | ResNet50 | Nasty | 94.28 | | 80.34($\pm$1.19) | **86.14**($\pm$0.01) | +5.80 |
| | | Normal | 94.9 | | 90.54($\pm$1.16) | **91.93**($\pm$0.04) | +1.39 |
| Without AT loss (black-box) | | | | | | | |
| CIFAR -10 | ResNet50 | Nasty | 94.28 | ResNet18 | 20.95($\pm$0.91) | **79.93**($\pm$0.28) | **+58.93** |
| | | Normal | 94.9 | | 22.08($\pm$0.56) | **80.71**($\pm$0.6) | +58.63 |

performance can improve up to $5.8\%$ and $58.93\%$, with grey-box and black-box teacher assumptions, respectively. These results clearly show that skeptical students can largely diminish the KD-immunity of a nasty model under even the data-free scenario.

### 5.6 Transferability of Nastiness on Skeptical Students

Similar to the transferability test on normal students (see Table 1), we also explored the transferability of a nasty teacher to a skeptical student. For this experiment, we use a skeptical student trained from a nasty teacher (ResNet50) as a teacher for a secondary student on CIFAR-100. Here, we used a normal ResNet18 as the secondary student model. Interestingly, Table 5 shows the performance of the secondary student improves by $1.67\%$ compared to the baseline ResNet18, following the same trend as a student distilled from a normal teacher. From these results, we conclude that, *a skeptical student not only reduces the nastiness of a teacher on its own performance, but also breaks the chain of transferability of nastiness to a secondary student*.

Table 5: Performance of a skeptical student (ResNet18) under transferability test on CIFAR-100.

| Teacher | Teacher type | Teacher Acc % | Student Acc % | $\Delta_{base}$ |
|---|---|---|---|---|
| ResNet50 | Nasty | 76.57 | 77.43 | -0.12 |
| ResNet18 | Nasty-distilled | 77.43 | 79.22 | +1.67 |
| ResNet50 | Normal | 78.04 | 78.90 | +1.35 |
| ResNet18 | Normal-distilled | 78.90 | 79.92 | +2.37 |

## 6 Conclusions

In this paper we presented a skeptical student who leverages a simple yet effective hybrid distillation strategy to diminish the effect of a nasty teacher and largely retain its classification performance. In particular, our experimental results showed that, when distilling from a nasty teacher, the performance of skeptical students is up to $\sim59.5\%$ higher than that of normal students. Our models also retain a similar performance as the normal student when distilled from a normal teacher, showing the general efficacy of the proposed KD under both nasty and normal teacher scenarios.

## 7 Broader Impact

The diminishing immunity of a nasty teacher for the proposed distillation technique highlights the growing concern of model IP protection in today's DNN-driven world. We believe this study will help the community understanding the limitations of such model IP protection techniques. Moreover, we also hope this will motivate the community to further study the fundamental limits of maintaining model confidentiality given access to a black-box teacher.

## 8 Acknowledgment

This work was supported in parts by NSF including grant number 1763747.

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
