# Supplementary Material

**Souvik Kundu, Qirui Sun, Yao Fu, Massoud Pedram, Peter A. Beerel**
Electrical and Computer Engineering
University of Southern California
Los Angeles, CA 90089
{souvikku, qiruisun, yaof, pedram, pabeerel}@usc.edu

## 1   Model Details

We used popular ResNet models viz. ResNet18 and ResNet50 models and MobileNetV2 as well as their skeptical variants to evaluate the efficacy of proposed hybrid distillation scheme. In particular, Table 1 provides the parameters and FLOPs details[1] of the corresponding normal and skeptical student variants. For the ResNet models we added the auxiliary classifiers (ACs) enhancements after the basic block layer number 2 and 3. For the MobileNetV2 variant these ACs are placed after stage 4 and stage 6 where a stage is a combination of linear bottleneck layers as defined by [1]. It is noteworthy that the skeptical models are more parameter-heavy due to the added parameter cost for the ACs. However, **during inference, we pay these added parameter and compute cost only when we use the ACs for ensemble classification**, which is a choice that we may opt out.

Table 1: Model parameters and FLOPs details

.

|         | **Model**        | ResNet18 | ResNet50 | MobileNetV2 |
|---------|------------------|----------|----------|-------------|
| Normal  | # **Parameters** | 11.2 M   | 23.5 M   | 2.3 M       |
|         | # **FLOPs (MACs)** | 0.56 G | 1.3 G    | 0.08 G      |
|         | **Model**        | ResNet18 | ResNet50 | MobileNetV2 |
| Skeptical | # **Parameters** | 11.8 M | 33.4 M   | 5.5 M       |
|         | # **FLOPs (MACs)** | 0.57 G | 1.53 G   | 0.15 G      |

## 2   Ablation Study with the Loss Components

As described in Section 4 of the original manuscript, the loss component of the student for distillation under data-available scenario has three major components: 1. $\mathcal{L}_{\mathcal{CE}}$, 2. $\mathcal{L}_T$, and 3. $\mathcal{L}_{SD}$. In this section we analyze the contribution of the self-distillation loss component. In particular, Table 2 shows the skeptical students' performance with only $\mathcal{L}_T$ and with both $\mathcal{L}_T$ and $\mathcal{L}_{SD}$ (along with the $\mathcal{L}_{\mathcal{CE}}$). For both CIFAR-10 and CIFAR-100 datasets, the skeptical students with proposed hybrid loss perform better compared to their counterparts with only one distillation loss component, $\mathcal{L}_T$, thus depicting the contribution of the self-distillation loss component.

## 3   More Results

Fig. 1 shows the performance of a student model upon distillation from a nasty teacher under various % of limited training data availability, fir CIFAR-100 dataset. Similar to that on CIFAR-10 dataset, the figure shows under different % of training data availability the skeptical student performs consistently better than its normal variant upon distillation from a nasty teacher.

---

[1]Evaluated for CIFAR-10 classification.

Table 2: Performance ablation study of skeptical students with various loss components, when distilled from a nasty teacher.

| Dataset | $\Phi_T$ | $\Phi_T$ Acc. (%) | $\Phi_S$ | Student Acc. (%) | | |
|---|---|---|---|---|---|---|
| | | | | Normal | Skeptical | |
| | | | | | $\mathcal{L}_{\mathcal{CE}}+\mathcal{L}_T$ | $\mathcal{L}_{\mathcal{CE}}+\mathcal{L}_T+\mathcal{L}_{SD}$ |
| CIFAR -10 | ResNet18 | 94.67 | ResNet18 | 94.13 | 94.8 | **95.09** |
| | | | ResNet18 | 94.38 | 95.0 | **95.16** |
| | ResNet50 | 94.28 | ResNet50 | 94.21 | 95.1 | **95.48** |
| CIFAR -100 | ResNet18 | 77.55 | ResNet18 | 75.0 | 76.5 | **77.33** |
| | | | ResNet18 | 72.28 | 75.4 | **77.25** |
| | ResNet50 | 76.57 | ResNet50 | 74.14 | 76.2 | **78.65** |

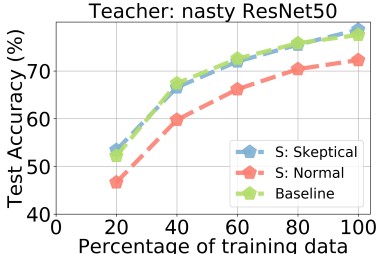

Figure 1: ResNet18 on CIFAR-100 dataset under different percentage of limited training data.

Fig. 2 shows examples of output logit distribution when both the student and teacher models have same architecture (i.e. DtoS). For DtoS scenario, the skeptical student (ResNet50) shows similar logit distribution as that of a normal teacher model and largely mitigates the multi peak logit distribution of the nasty teacher model.

## References

[1] Mark Sandler, Andrew Howard, Menglong Zhu, Andrey Zhmoginov, and Liang-Chieh Chen. MobileNetV2: Inverted residuals and linear bottlenecks. In *Proceedings of the IEEE conference on computer vision and pattern recognition*, pages 4510–4520, 2018.

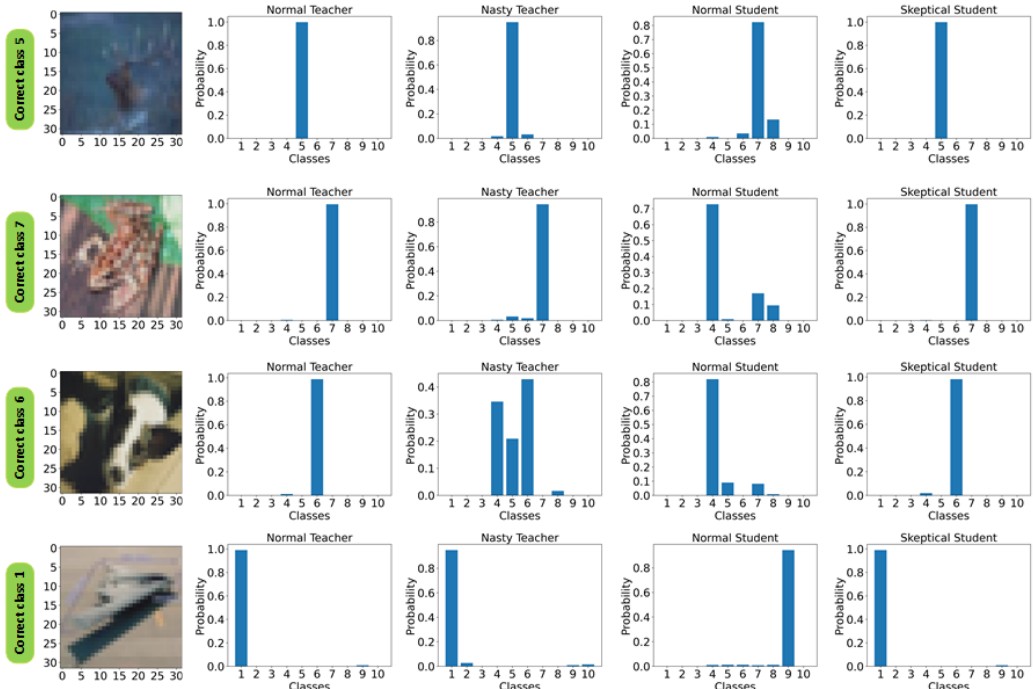

Figure 2: Logit response visualization after the softmax layer for distill to self scenario. Each row contains an example image from CIFAR-10 dataset and corresponding response for normal teacher, nasty teacher, normal student and skeptical student. Here, both the teacher and student models are chosen to be ResNet50.