# OpenReview forum: "Analyzing the Confidentiality of Undistillable Teachers in Knowledge Distillation"
_NeurIPS.cc/2021/Conference — NeurIPS 2021 Poster_

### Official Review · Reviewer_ELaj · 2021-07-10

**Rating:** 7
**Confidence:** 3

**Summary:**

The paper introduces a skeptical student distillation framework that diminishes the effect of a nasty teacher. By placing auxiliary classifiers at different depths of the student model to form a shallow subsection and introducing a new self-distillation term, the resulting skeptical student clearly outperforms a vanilla student in all data-available, data-limited, and data-free settings when learning from a nasty teacher. The proposed KD method (especially the ensembled outcome) also provides similar performance as a vanilla student when learning from a normal teacher, thus being applicable in most cases. Quantitative analysis and visualizations are provided to show that the method works as expected.


**Ethical Concerns:**

Raise safety or security concerns: the proposed technique could be potentially used to extract model IP from black-box ML service. It diminishes the effect of existing protection approaches, which is clearly acknowledged by the author.


**Limitations And Societal Impact:**

Yes, the author clearly addressed them in the paper.


**Main Review:**

Originality: how to effectively use KD to learn from a protected nasty teacher model is an interesting new question. The self-distillation technique is not new but is applied in the new setting. Related works are well cited.

Quality: The submission is technically sound and most claims are supported by empirical results. The work is complete and the comments for the evaluation experiments are fair.

Clarity: The paper is clearly written and should be detailed enough for reproducing the result. However, some major prior works (e.g. self-distillation network structure, details about using auxiliary classifier) are not clearly discussed and more details on those preliminaries can help readers to better understand the topic without jumping through references.

Significance: The paper discusses an interesting technique that to some extent compromises an existing model IP protection mechanism. There might be follow-up works on how to defend against such skeptical student model stealers. As also mentioned by the authors, the paper might motivate further investigations on model privacy protection.


**Time Spent Reviewing:**

3

---

> ### Author Response · Authors · 2021-08-09
> **Thank you for your insightful reviews and accepting our work**
>
> We thank the reviewer for the comments and for identifying the significant contribution of our work. We also thank the reviewer for recommending acceptance. We reference the reviewer with identifier ELaj as R4. Comment n of reviewer m is denoted as RmCn.
>
> **[R4C1]** We thank the reviewer for recognizing our work as moderately original.
>
> **[R4C2]** We also thank the reviewer for identifying the quality of our work to be technically and empirically sound.
>
> **[R4C3]** We will happily elaborate on the major prior works in the related work section, and provide details of the corresponding network structures in the experimental results.
>
> **[R4C4]** We appreciate the reviewer identifying the significance and impact of our work in the context of model IP protection.

---

> ### Author Response · Authors · 2021-08-26
> **Post-rebuttal discussion**
>
> Dear reviewer ELaj,
>
> We really appreciate your effort in providing insightful comments and honor your evaluation of our paper's novelty and contribution. We are glad to receive an "accept" from you. We will be happy to provide further clarification in case if you need any.
>
> Regards,
>
> Authors.

---

> > ### Comment · Reviewer_ELaj · 2021-08-30
> > **post-rebuttal discussion**
> >
> > I have read the author feedback and have taken it into my consideration

---

### Official Review · Reviewer_mcZU · 2021-07-15

**Rating:** 7
**Confidence:** 4

**Summary:**

The authors investigate the knowledge distillation immunity of Nasty teachers. A nasty teacher is a model that maintains good prediction performances but significantly degrades the performances of student models that distill knowledge from it. In this paper, the authors proposed a hybrid distillation that neutralizes the protection of nasty teachers, allowing the training of accurate student models.

**Limitations And Societal Impact:**

Yes

**Main Review:**

Overall, the paper is addressing an important problem and the experiments conducted by the authors demonstrated that the proposed approach can effectively bypass the immunity of nasty teachers.

My main concern with the paper is that the ability to steal an IP from the nasty teacher is solely evaluated with the accuracy. However, accuracy can be misleading due to predictive multiplicity. That is, both teacher and student models can exhibit high accuracy while having different properties. So it would be interesting to report more details on the difference between students distilled from normal teachers and those distilled from nasty teachers.

Minor remark:
The word "privacy" in the title can be misleading as it can give the impression that the paper is about the robustness of nasty teachers against privacy inference attacks, which is not the case.

Section 2 has some readability issues and will benefit from a proofreading.

**Time Spent Reviewing:**

7

---

> ### Author Response · Authors · 2021-08-09
> **Thank you for your insightful reviews and accepting our work**
>
> We thank the reviewer for the comments and for identifying the significant contribution of our work. We also thank the reviewer for recommending acceptance. We reference the reviewer with identifier mcZU as R3. Comment n of reviewer m is denoted as RmCn.
>
> **[R3C1]** Thank you for the nice summary of our work. In particular, we appreciate the reviewer's effort in identifying the motivation of this work as an important problem.
>
>  **[R3C2]** We appreciate the reviewer’s concern that our evaluation of the distilled models was only in terms of their classification accuracy. However, we would like to emphasize that the stealer’s objective is to successfully replicate the model’s classification performance, as mentioned in earlier literature [1]. In particular, the stealer need not obtain a teacher’s structure or weights because their goal is solely to use the teacher as a means of teaching a student (often less complex). The purpose of an undistillable teacher is to prevent the effective distillation to a student and this has been in prior work measured by the accuracy of the distilled student.
> However, we understand the concern of predictive multiplicity and thus would be happy to incorporate further performance details, including reporting the number of images successfully classified by students from normal teachers, but not from nasty, and similar insightful observations. In particular, Table R3T1 Column 2 shows the number of distinct test image samples that are only classified correctly by a student when it distills from a normal $\Phi_T$ and fails to classify correctly when distilled from a nasty $\Phi_T$. Column 3 shows the opposite scenario, meaning the exclusive image sample counts that the student can only classify correctly when it is distilled from a nasty $\Phi_T$. The large difference between these two numbers ensures a classification performance gap for a normal student. Whereas a small gap for skeptical students hints at the fact that it performs nearly similarly irrespective of the teacher type. We used ResNet18 and ResNet50 as student and teacher models, respectively, and evaluated on CIFAR-100.
> #### R3T1
> | Student type | Only from normal $\Phi_T$ | Only from nasty $\Phi_T$ |
> |--------|--------|:-----------:|
> | Normal | 1181 | 438 |
> | Skeptical | 666 | 577  |
>
> We will be glad to incorporate these insightful analyses into the final paper.
>
>  **[R3C3]** We understand the reviewer’s valid concern about the possible confusion of the word privacy in our title with the notion of privacy inference attacks. If we are allowed to change the paper title, we would be happy to replace the word “privacy” with “confidentiality” as recommended in earlier literature [2].
>
> [1] Undistillable: Making A Nasty Teacher That CANNOT teach students, ICLR 2021.
> [2] Defending Against Model Stealing Attacks with Adaptive Misinformation, CVPR 2020.

---

> > ### Comment · Reviewer_mcZU · 2021-08-18
> > **Thank you for the clarifications. Please consider updating the paper with the changes you proposed**
> >
> > Thank you for the clarifications. Assuming that you update the paper as you suggested, I will increase my score.

---

> > > ### Author Response · Authors · 2021-08-26
> > > **Post-rebuttal discussion**
> > >
> > > Dear reviewer mcZU,
> > >
> > > We really appreciate your effort in providing insightful comments and honor your evaluation of our paper's novelty and contribution. We are glad that you have provided an increased score of 7 after analyzing our rebuttal comments. We remain committed to incorporate the rebuttal comments in the final manuscript as suggested by you. We will be happy to provide further clarification in case if you need any.
> > >
> > > Regards,
> > >
> > > Authors.

---

### Official Review · Reviewer_NwTU · 2021-07-16

**Rating:** 6
**Confidence:** 4

**Summary:**

Knowledge distillation (KD) can be used by an adversary to obtain sensitive information from a teacher model presenting privacy risks. Nasty teachers were proposed to handle this problem where students were unable to learn or distill knowledge from nasty teachers. In this paper, the authors show that skeptical students using a combination of KD to intermediate shallow subsections of their model together with Self Distillation (SD) can extract information from a nasty teacher and perform well on classification tasks.

**Limitations And Societal Impact:**

Yes

**Main Review:**

The paper provides an attack against the nasty teacher-based defense proposed in prior work.

Some of my concerns about the work are as follows:
a. The authors motivate the idea using knowledge leakage and then introduce nasty teacher as a proposed defense. However, the paper does not provide any explicit threat model corresponding to the defense. It is important to assess if the adversary is able to extract the information while adhering to the constraints of the threat model.

b. Along similar lines, the metric used for showing the effectiveness of the attack is the classification accuracy. However, it is not clear what the exact leakage is due to the distillation. There is a need to decouple the two and show that indeed some sensitive information is being leaked.

c. In Table 2, when a normal student is used, the drop in accuracy from the baseline is not very significant (other than when MobileNetV2 model is used for CIFAR-100 and Tiny-ImageNet). This indicates two problems: First, the baseline protection mechanism using a nasty teacher is itself not very strong. A normal student is able to still distill most of the information from the nasty teacher. Second, for models where we observe a drop in accuracy, I think there is a need to isolate the reason - is it due to the nasty teacher or the model architecture being used? The authors report improved performance with skeptical students, but from a privacy standpoint, I am not convinced this really means much. The system seems to be already leaking a lot of information. This further goes back to the point made in (b) above.

d. I really like the use of self-distillation in the data-free distillation formulation. However, the improvement in test accuracy is not very significant as reported in Fig. 8. I agree that the gap is a little larger when only 20% of training data are available. Maybe, representing in tabular form will help bring out the improvement.

Writing Issues:
The paper is well-written. Some minor issues are pointed below:
Line 126-127: The sentence, "One the other hand.." is unclear. Moreover, this form of sentence structure is used to contrast with another observation. There is no such contrasting observation presented.

Line 148: The notation used in Eqn (2) is confusing. How is \Phi_{S}^{j} and  \Phi_{S}^{'} related?

**Time Spent Reviewing:**

8

---

> ### Author Response · Authors · 2021-08-09
> **Thank you for the insightful reviews**
>
> We thank the reviewer for the comments and reference reviewer with identifier NwTU as R2. Comment n of reviewer m is denoted as RmCn.
>
> **[R2Ca]** Thank you for pointing out that our threat models should be more explicit and justified. Our threat model has two dimensions. The first is the amount of training data being available, and we consider the full range of scenarios where none, some, or all of the training data is available to the attacker. The second dimension is the amount of knowledge the attacker has of the model and we consider both black-box and grey-box models (in which part of the activation maps but no weights are available to the attacker). In our response to [R1C2a] and [R1C4] (of reviewer id: j2bi), we expanded upon potential real-world justification of various combinations of these dimensions used throughout the paper as well as referenced prior work that have analyzed similar scenarios. In particular, we explored black-box models under data available scenarios and grey-box models under the data free scenario. We also added results for black-box models under the data-free scenario in Table R1T1 above (in [R1C4] of reviewer id: j2bi).
>
> **[R2Cb]** We thank the reviewer for pointing out that the information protected by nasty teachers and leaked via skeptical students is unclear. In fact, the specific information leaked is amorphous because what is being protected is the model performance.  In particular, the goal of the model developer is to prevent the stealer from using their model to develop a different model (perhaps with lower complexity) that has similar performance [1-4]. Hence, similar to earlier work [3], the performance of the stealer in terms of classification accuracy is the standard means of measuring the success of the attack. In this paper, we are showing that distillation using a skeptical student can be a successful attack even with undistillable teachers and will be happy to clarify this point in the final paper.
>
> **[R2Cc]** We appreciate the reviewer’s concern that the nasty teacher often can force a normal student to only have a 2-6% drop in accuracy (from a baseline of 77%+ accuracy). The reviewer questioned whether this level of protection is significant. Here, we would like to highlight that for safety-critical applications, like autonomous vehicles or drones, a drop in accuracy of even 1% may be considered significant.  Secondly, by 2025 around 75% of intelligent services may be at the edge [5] and thus largely resource-constrained.  For such edge deployments, small-scale models like MobileNet V2 are a more realistic target student models and the relative observed drop in accuracy compared to its baseline is much larger (up to ~60%). This may explain the high interest in undistillable teachers.
>
> The reviewer also rightly asked whether the drop in accuracy is more of a function of the undistillability of the nasty teacher or the (sometimes limited) architecture of the student model.
> We have indeed seen the performance of a student is largely a function of it’s own architecture (Table 2 and 3 Column 6 results). However, at the same time, this data shows that the nasty teacher can successfully degrade the student’s performance irrespective of its model architecture. Note that [3] has suggested that the small compute-efficient models suffer more from such teachers which may be linked to their limited learning capacity. We will happily elaborate these points in the final manuscript.
>
> Lastly, the reviewer was concerned that the undistillable teachers were already leaking a lot of information. As mentioned above, low-complexity student models such as MobileNet V2 are examples of where undistillable teachers appeared to be a very successful case of protecting model IP (from normal students).  However, our skeptical student’s significant performance gain by up to 59.5% shows that it can almost completely nullify the effect of an undistillable teacher.  Our results, in fact, thus raise a fundamental question as to if any model can sufficiently protect its performance from distillation-based attacks.
>
> **[R2Cd]** We really appreciate the fact that the reviewer liked our approach and in particular the novel use of self-distillation in the data-free scenario. Moreover, we will happily include the tabular representation of Fig. 8 as recommended by the reviewer. In particular, table R2T1 clearly shows the consistent superior performance of Skeptical students upon distillation from a nasty teacher (model: ResNet50) on both CIFAR-10 and CIFAR-100 (with various % of training data available scenario).
>
> #### R2T1
> | Dataset| $\Phi_S$ |   20%   |  40 % | 60%  | 80%  | 100% |
> |--------|--------|:-----------:|------:|-------------:|-------:|---------:|
> | CIFAR-10 | ResNet18(Normal) |  84.59 | 90.19 | 92.41 | 93.6 | 94.38 |
> | CIFAR-10 | ResNet18(Skeptical)  | **88.23** |  **92.06** | **93.79** | **94.44** | **95.16** |
> | CIFAR-100 | ResNet18(Normal) |  46.6 | 59.7 | 65.67 | 70.38 | 72.28 |
> | CIFAR-100 | ResNet18(Skeptical)  | **53.42** |  **66.6** | **72.5** | **75.5** | **77.25** |
>
>
> **[R2Ce]** We thank the reviewer for appreciating the paper’s writing. We will update the sentence in L 126-127 to remove the phrase “On the other hand”, as recommended by the reviewer.
>
> **[R2Cf]**  We apologize for the confusing notation used in L148. $\Phi_S’$ represents the shallow sub-section of the student model $\Phi_S$. We will clarify that this shallow sub-section starts from the initial layer and ends at some intermediate layer-block of the student, and we assume this shallow sub-section is composed of multiple layer-blocks. A layer-box is made of multiple convolutional layers, e.g. basic-block for ResNet18. We transfer KD loss from a teacher to the skeptical student at the AC located at the end of the last layer block of $\Phi_S’$ which is denoted as $i_{\Phi_S’}$. Now if the student model has total N layer-blocks, we only choose a  layer-block that is present after $\Phi_S’$ and place another AC to transfer self-distillation loss. So, in our case, ${\Phi_j}$ is a shallow-sub section of the student that also starts from the initial layer but essentially ends at a layer-block which is after the one denoted as $i_{\Phi_S’}$.
>
> [1] Defending Against Model Stealing Attacks with Adaptive Misinformation, CVPR 2020.
> [2] Prediction Poisoning: Towards Defenses Against DNN Model Stealing Attacks, ICLR 2020.
> [3] Undistillable: Making A Nasty Teacher That CANNOT teach students, ICLR 2021.
> [4] PRADA: Protecting Against DNN Model Stealing Attacks, Euro S & P 2019.
> [5] https://www.gartner.com/smarterwithgartner/what-edge-computing-means-for-infrastructure-and-operations-leaders/

---

> > ### Comment · Reviewer_NwTU · 2021-08-30
> > **Post-rebuttal comments**
> >
> > I would like to thank the authors for the detailed clarifications. I have read the clarifications presented for my comments as well as those for reviewer j2bi (especially the ones related to motivation). I understand and agree with response R2Cb. However, I am not fully convinced about R2Cc and R2Cd. I really think this attack does not mean much especially because the nasty teacher defense is terrible against most models. While I appreciate the need to protect against any significant drop in model accuracy, I am not certain that for the safety critical applications, used as motivation, we will use a MobileNet V2 model. Furthermore, the decoupling of the impact of model capacity and nasty teacher is not clear to me. In view of the above, I think that the paper continues to remain slightly below the NeurIPS bar.

---

> > > ### Author Response · Authors · 2021-08-30
> > > **Rolling rebuttal clarifications**
> > >
> > > Dear Reviewer NwTU,
> > >
> > > We are glad that our initial rebuttal partly clarified a few of your concerns. Here, we are focusing on the parts (primarily three points) where we recon more justification is necessary.
> > >
> > > ***1. Nasty teacher defense is terrible against most models***
> > >
> > >  We agree that the nasty teacher does not always significantly drop the stealer's performance, however, we would like to highlight that on many models the defense is not terrible. As highlighted in the following table (part of Table 2 and 4 in our manuscript), the nasty teacher can provide a defense (because the drop in performance is significant) even when applied to larger models like ResNet18/ResNet50.  In particular, under the full data-available, 60% data-available, and data-free scenario, they can significantly degrade the stealer's performance by up to 9.47%, 11.88%, and 10.2%, respectively. It is important to note here that earlier defense mechanisms have assumed the attacker has only access to synthetic data. Whereas the nasty teacher assumed no, part or even full data is available.  We have also justified such assumptions with real-life scenarios in R1C2a and R1C4.
> > >
> > > | Dataset | $\Phi_T$ (potential model IP)|      $\Phi_S$ (potential stealer)  | $\Phi_S$  Acc % | $\Delta_{base}$ Acc % |
> > > |----------|:-------------:|------:|-------------:|-------:|
> > > |CIFAR100 (100% data-available)| ResNet18 | ResNet18 | 75.00 | -2.55|
> > > |CIFAR100 (100% data-available)| ResNet50 | ResNet18 | 72.28 | -5.27|
> > > |CIFAR100 (100% data-available)| ResNet50 | ResNet50 | 74.14 | -3.9|
> > > |CIFAR100 (60% data-available)| ResNet50 | ResNet18 | 65.67 | -11.88|
> > > |Tiny-ImageNet (100% data-available)| ResNet18 | ResNet18 | 53.60 | -9.47|
> > > |CIFAR10 (data-free)| ResNet50 | ResNet18 | 80.34 | -10.2|
> > >
> > > We would like to reiterate the fact that replicating a model's performance can truly nullify the need for the original model and thus significantly degrade the value of the model’s IP. Moreover, the simplicity of the distillation threat model has made IP protection against such attacks particularly attractive. This is why we think, the nasty teacher’s promise of protecting against model stealers (**by making sure the stealer always has degraded performance**) led to the paper becoming a spot-light ICLR 2021 paper. Thus, many researchers may now believe in this promise.
> > >
> > > Our paper essentially shows that this promise can be broken. Given the attention earlier conference has given to this promise, we believe the fact that it can be broken has significant merit. More precisely, our **proposed skeptical students consistently provide superior performance irrespective of the strength of the nasty teacher defense**. In particular, we have shown in Table R1T1 that skeptical students achieve superior performance of up to 58.98% when distilled from a black-box teacher without any training data.
> > >
> > > ***2. Not certain that for the safety-critical applications, used as motivation, we will use a MobileNet V2 model***
> > >
> > > Here, we would like to highlight the fact that MobileNetV2 is widely used in various safety-critical applications [2-7] because many of these applications also have limited resource requirements and require fast inference speed. Autonomous cars and drones must balance between the resource demands of ML and vehicle range [1-3].  Health-care applications are increasingly on mobile devices in which optimizing battery lifetime is essential making the use of large models impractical [4-7]. The tradeoff between speed and accuracy is well documented [8].  For these applications, MobileNetV2 is representative of the smaller models needed, as documented by MNCs like Google and Amazon [9, 10]. For example, the use of MobileNetV2 as the backbone network in object detection compared to a larger alternative can increase the FPS by up to 35% [1]. We will happily add these details to highlight wide-spread applications of MobileNetV2 type models.
> > >
> > > ***3. Decoupling of the impact of model capacity and nasty teacher is not clear***
> > >
> > > The original undistillable paper [11] showed the impact of a nasty teacher on models of various capacities and sizes. While showing significant degradations overall models, they concluded that their approach works best for smaller models. In contrast, we would like to emphasize that our skeptical students nullified the impact of the nasty teacher for all tested models that have a wide range of capacities. We will be happy to emphasize this in the revised manuscript.
> > >
> > > [1] https://analyticsindiamag.com/why-googles-mobilenetv2-is-a-revolutionary-next-gen-on-device-computer-vision-network/
> > >
> > > [2] http://diposit.ub.edu/dspace/bitstream/2445/131201/3/memoria.pdf
> > >
> > > [3]  "Object Detection from the Video Taken by Drone via Convolutional Neural Networks", Mathematical Problems in Engineering, 2020.
> > >
> > > [4] Polyp Segmentation in Colonoscopy Images using U-Net-MobileNetV2, arxiv 2021.
> > >
> > > [5] Extracting Possibly Representative COVID-19 Biomarkers from X-ray Images with Deep Learning Approach and Image Data Related to Pulmonary Diseases, Journal of Medical and Biological Engineering 2020.
> > >
> > > [6] An artificial intelligent diagnostic system on mobile Android terminals for cholelithiasis by lightweight convolutional neural network, PLOS ONE 2019.
> > >
> > > [7] MobileNetV2 Ensemble for Cervical Precancerous Lesions Classification,  MDPI 2020.
> > >
> > > [8] Speed/accuracy trade-offs for modern convolutional object detectors, CVPR 2017.
> > >
> > > [9] https://ai.googleblog.com/2018/04/mobilenetv2-next-generation-of-on.html
> > >
> > > [10] https://aws.amazon.com/marketplace/pp/prodview-lc2ud4oz3xjja
> > >
> > > [11] Undistillable: Making A Nasty Teacher That CANNOT teach students, ICLR 2021.

---

> > > > ### Comment · Reviewer_NwTU · 2021-08-30
> > > > **Final score**
> > > >
> > > > Thank you for the clarifications. I am increasing my score to 6.

---

> > > > > ### Author Response · Authors · 2021-08-31
> > > > > **Thank you for your post-rebuttal response**
> > > > >
> > > > > Dear reviewer NwTU,
> > > > >
> > > > > Thank you very much for your useful suggestions and for asking for important clarifications. We are glad to receive an "accept" from you and remain committed to incorporating all the valid clarifications that you have asked for and make the final manuscript a better one.
> > > > >
> > > > >
> > > > > Regards,
> > > > >
> > > > > Authors.

---

### Official Review · Reviewer_j2bi · 2021-07-17

**Rating:** 6
**Confidence:** 5

**Summary:**

1) The main contribution of this paper is a new knowledge distillation (KD) framework called "skeptical student" that can circumvent the "immunity (undistillability)" provided by a "nasty (defensive/secretive)" teacher.

The concept of "nasty" teacher was proposed in [A] to prevent stealing of a teacher model by a malicious student. This paper proposes a countermeasure that borrows ideas from self-distillation, where knowledge is transferred to shallow subsections of the student model, each with its own auxiliary classifier (AC). The proposed skeptical student performs knowledge distillation both to the shallow sub-sections and to the full student model to overcome the impact of a nasty teacher.

[A] Ma et al., "Undistillable: Making a nasty teacher that cannot teach students", ICLR 2021

2) The paper also includes an algorithm for training skeptical students in a data-free KD scenario (the original data used to train the teacher model is not available).

3) Experimental results on three datasets indicate that the proposed skeptical student can learn well both from the normal and nasty teachers.

**Ethical Concerns:**

Not applicable.

**Limitations And Societal Impact:**

The paper can be enhanced by including strong motivating examples to justify the need for this work.

**Main Review:**

1) The paper is moderately original. It uses existing concepts such as self-distillation and zero-shot knowledge transfer in a clever way to achieve the goal of training skeptical students that can learn even from defensive teachers.

2) The main problem with the paper is its motivation. Under what real-world circumstances are defensive teachers necessary, especially when the training data is assumed to be known? Even in the data-free scenario, an attacker who wishes to steal IP information of the target teacher model (including data used to train the model) can carry out direct model extraction attacks, rather than relying on the knowledge distillation framework. A simple and straightforward approach to mitigate such attacks is to add random noise to the softmax values (if they ever need to be released), rather than train a defensive teacher. If the need for defensive teachers itself is not strong enough, the motivation for skeptical students is very weak.

3) The key observation of the paper is that the proposed skeptical student can learn well from both normal and defensive teacher models. If that is the case, why not treat the skeptical student as a new knowledge distillation framework and analyze its properties more comprehensively purely from the knowledge distillation perspective (e.g., theoretical analysis on why this approach is better and/or experiments on more datasets to determine its strengths and limitations).

4) As explained earlier, if there is likely to be any use case for defensive teachers and skeptical students, it will be only under the data-free KD scenario. However, this scenario is treated very casually both in terms of explanation and experimental results. For example, the paper claims to include attention transfer loss, which is not at all possible unless one has white/gray-box access to the target teacher model. If someone has white/gray-box access to the teacher model, why would he/she attempt to steal the IP that is already available to him/her? Even with the attention transfer loss and access to 20% of the original training data, the proposed skeptical student performs only marginally better than the regular/normal student under the data-free scenario. This further illustrates the futility of this whole approach.

5) If my understanding of Figure 2 and Section 3.2 is correct, an auxiliary classifier trained with only level 1 features (shallow sub-section) of the student network works as well as the teacher model. In that case, why not simply go for a shallow student network with only level 1 features? In general, what is the impact of the depth of the student model on the proposed approach. In traditional knowledge distillation, one usually goes for shallower/simpler student architectures compared to the teacher. But if the goal is to steal the IP of the teacher, will going for shallower or deeper student networks help in any way?

6) The overall organization of the paper is very confusing and does not flow well. For example, what is the need for Section 3.1 (transferability impact of nasty teachers) if it is not used anywhere else in the paper.


**Time Spent Reviewing:**

2

---

> ### Author Response · Authors · 2021-08-09
> **Thank you for the insightful reviews**
>
> We thank the reviewer for the comments and reference reviewer with identifiers j2bi as R1. Comment *n* of reviewer *m* is denoted as R*m*C*n*.
>
> **[R1C1]**  We thank the reviewer for appreciating the originality and cleverness of our skeptical student models and the care we put into their evaluation. We happily clarify the concerns posed by the reviewer below.
>
> **[R1C2a]** We appreciate the reviewer’s concern regarding the lack of explicit motivation of our work and particularly the desire for real-world examples of defensive teachers where training data is available to an attacker. We think it is clear that protecting the performance of the models deployed for AI-enabled services (e.g.: home assistants, autonomous vehicles, commercial cloud APIs) is important due to the large effort invested in model training. We also agree with the reviewer’s insightful concern that training data might *not be intentionally* released by the same service providers that wish to protect their model IP (victim). However, there are scenarios where data can still become available: 1) The model trainer has bought training data from a separate company which might sell the same data to others; 2) The training data is leaked either from within their organization or from another organization that has access to the data (e.g., [7]). For both cases, the model IP provider may want to protect their model from a potential data leak. In particular, model IP providers may wish to prevent stealers from obtaining similar performance as their victim model with a more compute-efficient replacement model [8].
> Moreover, there are open-source large-scale datasets (e.g., the traffic dataset BDD100K [9] released by UC Berkeley) that can effectively act like the original training data set in a full or limited data scenario. Although not explicitly stated, we believe these situations are the underlying motivation for references that have proposed full and limited data scenarios [1, 6].
>
> **[R1C2b]** We agree with the reviewer that we should reference direct model extraction attacks. The main difference with model extraction attacks is that they often require information of the victim model architecture [2] which might not be available in black-box scenarios and rely on hyper-parameter searching [2]. In contrast, knowledge distillation-based performance replication is effective between models of different architectures [4], mitigating the need for knowledge of the victim model architecture, and never requiring hyper-parameter search [1, 3].  Thus, while both attacks are important, it is clear that KD-based attacks deserve attention.
>
> **[R1C2c]** We appreciate the reviewer’s alternate suggestion to incorporate softmax random noise to fool a model stealer. We point out that, in fact, making a teacher undistillable effectively adds noise to the softmax outputs. This can be seen in Fig. 5 of our paper where we show that non-correct classes often have non-negligible values. The uniqueness of nasty teachers is that this noise is not randomly added, but added in a way that introduces a false sense of generalization that does not impact performance.  We believe for this reason understanding the limitation of the nasty teacher and in particular, the proposal of the skeptical students is an important contribution.
>
> We will happily incorporate these points in the final version of our paper, if accepted.
>
> **[R1C3]** We appreciate the reviewer’s point that skeptical students should be viewed as a new generic KD model and studied more comprehensively, including additional theoretical analysis. We would like to note that it is for this reason that we have studied the advantages of skeptical students on both nasty and normal teachers and will happily emphasize the importance of additional theoretical analysis in our future research.
>
> However, in light of the spotlight nature of the 2021 undistillable teacher paper, we hope to have a more immediate impact by focusing on their trustworthiness and the role skeptical students can have on thwarting this form of IP defense. In particular, we believe this paper has three significant contributions towards this goal: 1) Analyzes the limitations of a nasty teacher’s ability to transfer its nastiness to student models; 2) Proposes skeptical students' under both data available (full and limited) and data-free scenarios as a means of thwarting the potential nastiness of a teacher; and 3) Empirically shows that skeptical students consistently outperform conventional KD-based models both under distillation from a nasty teacher. Thus, this paper raises a fundamental question of model protection in distillation frameworks, including both limited data and data-free scenarios.
>
> Lastly, we note that we have also validated the performance of skeptical students on the SVHN dataset. However, because they yielded similar results as the similarly complex datasets tested, we did not include them in the paper. We have left the analysis on ImageNet for future research as it is known that even basic KD approaches have challenges on this dataset [5] and we believe future improvements to KD techniques is a complementary research direction.
>
> **[R1C4]** We thank the reviewer for pointing out that our analysis of skeptical students using KD under data-free scenarios must justify how the attacker may have access to the intermediate layer activation maps but not the model weights. This assumption is based on the realization that hardware/software security always comes at a cost and designs in which the model weights are stored in secure memories but the activations are subject to physical or side-channel attacks are reasonable. This means the model acts as a grey-box as we and other literature [1] have assumed in the data-free scenario. We would also like to emphasize that this paper *never* assumes the model is a white-box.
>
> However, we agree that a black-box model (with no attention transfer loss) under the data-free scenario may be more likely and thus present the corresponding results in Table R1T1.
> #### R1T1:
> | $\Phi_T$|      $\Phi_S$      |  $\Phi_T$ Acc % | $\Phi_T$  black-box | Data-free|  AT-loss | $\Phi_S$ Acc % | $\Delta$ Acc % |
> |----------|:-------------:|------:|-------------:|-------:|---------:|------------:|--------:|
> | ResNet50 (Normal) |  ResNet18 | 94.9 | Yes | Yes | No | 22.08 | -- |
> | ResNet50 (Normal) | ResNet18 Skeptical |    94.9 |  Yes | Yes | No | **80.71** |  58.63 |
> | ResNet50 (Nasty) |   ResNet18   |   94.28 | Yes | Yes | No | 20.95 | -- |
> | ResNet50 (Nasty) |   ResNet18 Skeptical  |   94.28 | Yes | Yes | No | **79.93** | 58.98 |
>
> Notice that the proposed skeptical students outperform conventional KD-based students while distilling w/o attention-transfer (AT) loss by up to  **58.98**%. These results highlight the impact of the proposed self-distillation in data-free KD and establish the superiority of the proposed method under black-box data-free scenarios.
>
> Finally, we would like to clarify that for all limited data experiments we have assumed the teacher model to be black-box and have *not* used AT-loss to train and evaluate the students’ performance. We commit to updating the data-free and limited data section with all these findings and detailed clarifications.
>
> **[R1C5]** It appears that our terminology caused some confusion. We confirm that we always keep the full student model, even when we transfer the knowledge to its shallow sub-section via an auxiliary classifier (AC) and have always reported only the accuracy at the student’s final classifier (C). We will clarify this point in the paper.  However, to expand upon this point here, Table R1T2 compares the accuracy after the shallow AC to that from after the final classifier. It confirms that  the AC’s performance is much lower than that of the final C irrespective of the depth of the AC.
> #### R1T2:
> | Nasty $\Phi_T$|    $\Phi_S$      |  $\Phi_T$ Acc % | $\Phi_T$  black-box | AC loc | Acc% at AC | $\Phi_S$  final classifier acc%
> |----------|:-----------:|------:|-------------:|-------:|---------:|--------:|
> | ResNet50 |  ResNet18 | 76.57 | Yes | 1 | 66.17 | 77.19
> | ResNet50  | ResNet18 |   76.57 |  Yes | 2 |  70.08 | 75.77
> | ResNet50 |   ResNet18   |  76.57 | Yes | 3 | 71.58 | 74.53
>
> Lastly, we also agree that often the student should be a model of lower complexity than the teacher and analyzed various such cases in Tables 2 and 3.
>
> **[R1C6]** We apologize for our confusing paper organization. Section 3.1 motivated the fact that a normal student distilled from a nasty teacher, acts as a nasty teacher for any downstream distillation task. This means it is unlikely that an attacker can steal the model performance via re-distillation from the student. To complement this analysis, we also evaluated our skeptical student's performance in the re-distillation scenario and showed that it *completely evade nastiness* and can act as a good teacher. This means there is potential performance leakage even from distilled models. However, we presented this result only in the supplemental material, which may be a source of some confusion. We will happily incorporate these results in the main manuscript to show the relevance of the analysis of Section 3.1.
>
> [1] Undistillable: Making A Nasty Teacher That CANNOT teach students, ICLR 2021.
> [2] PRADA: Protecting Against DNN Model Stealing Attacks, Euro S & P 2019.
> [3] Zero-shot Knowledge Transfer via Adversarial Belief Matching, NeurIPS 2019.
> [4] Contrastive Representation Distillation, ICLR 2020.
> [5] On the Efficacy of Knowledge Distillation, ICCV 2019.
> [6] Defending Against Model Stealing Attacks with Adaptive Misinformation, CVPR 2020.
> [7] https://www.nytimes.com/2018/04/04/us/politics/cambridge-analytica-scandal-fallout.html
> [8] Distilling the Knowledge in a Neural Network, NeurIPS 2015.
> [9] BDD100K: A Diverse Driving Dataset for Heterogeneous Multitask Learning, CVPR 2020.

---

> > ### Comment · Reviewer_j2bi · 2021-09-03
> > **Response to Author Rebuttal**
> >
> > My two main concerns in the original were related to the motivation and as well as its performance under the data-free KD scenario. While I'm still not completely convinced about the motivation aspect, the authors have done sufficient work (including some new results without attention transfer loss) to show that the proposed approach can work under the data-free KD scenario. Hence, I'm upgrading my overall score from 4 to 6.

---

> > > ### Author Response · Authors · 2021-09-03
> > > **Thank you**
> > >
> > > Dear reviewer j2bi,
> > >
> > > We are glad to receive these comments and an "accept" from you. As promised we will incorporate the additional results, rearrange the paper (bringing some results from supp. to original manuscript), and add additional motivations for the work.
> > >
> > > Regards,
> > >
> > > Authors

---

### Decision · Program_Chairs · 2021-09-27

**Decision:**

Accept (Poster)

**Comment:**

Previous work defined a *nasty teacher* to be a specially trained network that yields nearly the same performance as a normal one; but if used as a teacher model, it will significantly degrade the performance of student models that try to imitate it (with the goal of protecting against model stealers). In this paper, the authors proposed an attack that neutralizes the protection of nasty teachers, allowing the training of accurate student models. The reviewers agree that this is an interesting paper, all of them supporting acceptance.